# Predictive Factors of Deep Vein Thrombosis in Gynecologic Cancer Survivors with Lower Extremity Edema: A Single-Center and Retrospective Study

**DOI:** 10.3390/healthcare8010048

**Published:** 2020-02-27

**Authors:** Jungin Kim, Hyun-Jun Kim, Seunghun Park, Dong Kyu Kim, Tae Hee Kim

**Affiliations:** 1Department of Rehabilitation Medicine, Konkuk University Chungju Hospital, Chungju 27478, Korea; jsm00339@naver.com (J.K.); silveryluna@gmail.com (S.P.); peyous@hanmail.net (D.K.K.); 2Department of Obstetrics & Gynecology, School of Medicine, Konkuk University, Chungju 27478, Korea; icarus@kku.ac.kr; 3Research Institute of Medical Science, Konkuk University School of Medicine, Seoul 05029, Korea

**Keywords:** neoplasms, lower extremity, edema, venous thrombosis

## Abstract

This study was conducted to examine predictive factors of deep vein thrombosis (DVT) in gynecologic cancer survivors with lower extremity edema (LEE). In the current single-center, retrospective study, there was a total of 315 eligible patients, including 80 patients with DVT and 235 without DVT. They were therefore divided into two groups: the DVT group (*n* = 80) and the non-DVT group (*n* = 235). Then, baseline and clinical characteristics of the patients were compared between the two groups. In our study, distant organ metastasis, advanced stage, lymphadectomy, and amount of intraoperative blood loss had a positive predictive value for the occurrence of DVT in gynecologic cancer survivors presenting LEE. In conclusion, our results indicate that it is necessary to consider the possibility of LEE arising from DVT in gynecologic cancer survivors with advanced-stage cancer, distant organ metastasis, lymphadectomy, and intraoperative blood loss over 1500 mL.

## 1. Introduction

The relationship between venous thrombosis and malignancy was first described by Trousseau in 1865. Since then, it has been advocated by multiple clinical, pathologic, and laboratory studies [1,2]. According to Virchow, there is a triad of risk factors that contribute to venous thromboembolism; these include venous stasis, endothelial injury, and hypercoagulable states [3]. Patients with cancer are vulnerable to thrombosis arising from hematologic and biochemical abnormalities. For example, ovarian cancer cells are capable of forming and degrading thrombin. In addition, gynecologic malignancies are characterized by increased fibrinolytic activity. Furthermore, patients receiving surgery, chemotherapy, or radiotherapy are at increased risks of developing thrombosis [4,5].

According to epidemiological studies, patients with advanced-stage cancer are at increased risks of developing idiopathic venous thrombosis or thromboembolism. This deserves special attention [6].

Venous thromboembolism (VTE) is a leading cause of death in patients with cancer; it comprises deep vein thrombosis (DVT) and pulmonary embolism (PE) [7]. Its incidence of postoperative venous thromboembolism in ovarian cancer is reported to be 13.2% (DVT 71.6%, PE 24.3%, both 4.1%) [8]. In more detail, without prophylaxis, the incidence of VTE is estimated at approximately 10–40% [9]. With prophylaxis, it is estimated at 1.14% in patients diagnosed with gynecological disease, 0.7% in those undergoing laparoscopic gynecological surgery, 0.3% in those undergoing urogynecological surgery, and 4% in those with gynecological malignancies [10,11,12,13]. It remains problematic, however, that most published studies have evaluated symptomatic cases rather than asymptomatic ones as the latter could be frequently neglected without efficient methods for detecting VTE. Indeed, approximately 50% of total patients with VTE are presumed to be silent cases [14]. It can therefore be inferred that the actual incidence of postoperative VTE might be higher than compared to published reports [14]. 

Lower extremity edema (LEE) commonly occurs in patients with advanced-stage cancer, and it often poses a diagnostic dilemma for clinicians because of its non-specific symptoms. The diagnosis can be confirmed by performing a difference in circumference (>2 cm) and/or volume (>200 mL) between the affected and unaffected extremity. A variety of etiologic factors are involved in its pathogenesis, which include lymph blockage from tumor progression or anticancer treatment, DVT, hypoalbuminemia, renal or cardiac failure, and thyroid dysfunction [15]. Moreover, lymphedema is also considered the most likely cause of the LEE and this may obscure signs and symptoms of DVT [16]. It would therefore be mandatory to make a correct diagnosis of DVT, which is essential for lowering the risk of PE [17,18].

Various modalities are used to diagnose DVT. Of these, venography is considered a “gold standard”, but it has disadvantages, such as invasiveness, high cost, requirement of technical expertise, pain, unavailability for cases of allergy or renal insufficiency, difficulty of interpretation, and high inter- and intra-observer variability [19]. Alternative imaging modalities, such as computed tomography (CT) and magnetic resonance (MR) venography, are therefore used. But their applicability is also limited [19]. Currently, non-invasive diagnostic modalities are used to rule out suspected symptomatic DVT, and these include pretest probability estimation, D-dimer, and ultrasonography [19,20,21]. Moreover, compression vein ultrasonography with color Doppler flow or duplex ultrasonography is also recommended [22].

According to the guidelines from American College of Chest Physicians (ACCP) and American College of Obstetricians and Gynecologists (ACOG), appropriate preventive interventions are recommended based on diverse levels of postoperative risks of developing VTE [9,23]. Nevertheless, there is a paucity of evidence that advocates for the necessity to stratify patients undergoing gynecological surgery according to the level of risks of developing VTE [24,25,26]. In these patients, several postoperative risk factors of developing VTE have been suggested. These include body mass index (BMI) 30 or 40 kg/m^2^, operation time >180 minutes, cancer surgery, and blood transfusions of 2000 mL [19,27,28]. But this cannot be generalized, because the corresponding studies have failed to efficiently assess risk factors of developing VTE.

Given the above background, we conducted this single-center, retrospective study to examine predictive factors of the DVT. 

## 2. Materials and Methods

### 2.1. Study Patients and Setting

We analyzed a total of 580 patients (*n* = 580) with gynecologic cancer who had been treated at our medical institution between January 2012 and December 2018. 

Inclusion criteria for the current study were as follows:Gynecologic cancer survivors who are living with, through, and beyond cancer since the diagnosis of cancer, receiving continuous treatment, endeavoring to reduce the risk of recurrence, and managing chronic diseaseWomen with a swollen legWomen with confirmed radiological evidence of DVTWomen with metastasis to the lower extremitiesWomen with Eastern Cooperative Oncology Group (ECOG) performance status of 0 or 1 [29]

Exclusion criteria for the current study were as follows:Pregnant women (*n* = 0)Women receiving any perioperative prophylaxis or anti-coagulation therapies (*n* = 15)Women who were preoperatively diagnosed with DVT or PE (*n* = 10)Women who were lost-to-follow-up (*n* = 25)Women who are deemed to be ineligible for study participation according to our judgment (*n* = 0).

The current study was approved by the Institutional Review Board (IRB) of our medical institution (KUCH 2019-08-027).

### 2.2. Patient Evaluation and Criteria

In our series, we performed a retrospective review of the medical records and thereby analyzed baseline and clinical characteristics of the patients. These include age, the type of malignancy (*e.g.*, cervical cancer, endometrial cancer, and ovarian cancer), TNM stage at initial diagnosis, regional lymph node or distant organ metastasis, duration of disease (the time elapsing from the diagnosis of cancer to the evaluation of edema), BMI, co-morbidities (*e.g.*, diabetes mellitus, hypertension, hyperlipidemia, atrial fibrillation, heart failure, and rheumatic disease), treatment modalities (*e.g.*, surgery, chemotherapy, radiotherapy, and combination of more than two regimens), circumference of the lower extremity measured 10 cm above or below the upper border of the patella, and D-dimer levels. 

The anatomical location of the DVT was divided into proximal (the inferior vena cava [IVC], iliac, femoral, and popliteal veins) or distal (the anterior, posterior tibial, peroneal, and muscular veins).

Depending on the presence of DVT on CT venography, the patients were divided into two groups: the DVT group and the non-DVT group (Figure 1). Then, baseline and clinical characteristics of the patients were compared between the two groups.

To consider the possibility of lower extremity edema arising from DVT, a CT venography was performed for (A) the common femoral vein, (B) ① the superficial femoral vein ② the deep femoral vein, (C) the popliteal vein, and (D) the popliteal vein.

### 2.3. Statistical Analysis

All data was expressed as mean ± standard deviation. Statistical analysis was done using SPSS 18.0 for Windows (SPSS Inc., Chicago, IL, USA). To compare the baseline and clinical characteristics of the patients between the two groups, we performed a Mann-Whitney *U*-test. In addition, we also performed the χ^2^-test to identify the correlations between categorical variables and the incidence of DVT. Furthermore, univariate and multiple logistic regression analyses were also performed to identify significant correlations between risk factors of developing DVT and adjusted or unadjusted variables. Their results were expressed as odds ratios (ORs) with 95% confidence intervals (95% CIs). A *p*-value of <0.05 was considered statistically significant.

## 3. Results

### 3.1. Baseline Characteristics of the Patients

A total of 315 patients met inclusion/exclusion criteria. These include 80 patients with DVT and 235 without DVT. They were therefore divided into two groups: the DVT group (*n* = 80) and the non-DVT group (*n* = 235). The study flow chart is shown in Figure 2. In addition, baseline characteristics of the patients are represented in Table 1 and Table 2.

### 3.2. Location of the DVT

For the location of the DVT, there were 25 cases (31.3%) in the iliac vein, 20 cases (25.0%) in the femoral vein, 15 cases (18.8%) in the popliteal vein, 15 cases (18.8%) in the peroneal vein, F and 5 cases (6.3%) in the IVC.

### 3.3. Predictive Factors of the DVT

As shown in Table 3, the incidence of DVT had no significant correlation with the treatment modalities and co-morbidities. In addition, there were no significant differences in the circumference of the lower extremity, regional lymph node involvement, and D-dimer levels between the two groups. But distal organ metastasis, advanced-stage cancer, lymphadectomy, operation time ≥3 hours, and amount of intraoperative blood loss ≥1500 mL were significantly more prevalent in the DVT group as compared to the non-DVT group (*p* < 0.05).

### 3.4. Results of Univariate and Multivariate Analyses of Possible Predictive Factors

We performed both univariate and multivariate analyses of predictive factors, such as BMI, distant organ metastasis, advanced stage, lymphadectomy, operation time ≥3 hours, and amount of intraoperative blood loss ≥1500 mL, showing a significant difference between the DVT group and the non-DVT group. This showed that distant organ metastasis, advanced stage, lymphadectomy, and amount of intraoperative blood loss ≥1500 mL were found to be significant predictive factors (Table 4).

## 4. Discussion

VTE, presenting as DVT and PE, is a major cause of morbidity and mortality [30]. It is known that patients with malignancy are at a 6-fold greater risk of developing VTE and those with gynecologic cancer are at the greatest risk of developing VTE among all malignancies [31].

VTE is the second cause of mortality in patients with gynecologic cancer, and it has been reported that the risk of DVT and the incidence of PE were estimated at 17–40% and 1–26% in women undergoing gynecologic surgery [9,32].

It has been suggested that various risk factors are involved in cancer-related VTE [33]. Such risk factors are evaluated based on two well-known instruments. The Caprini risk assessment model (RAM), originally developed for surgical patients, aims to promote the derivation of risk factors of developing VTE. To do this, individual risk factors are summed and patients are divided into four categories accordingly: “low risk” (0–1 points), “moderate risk” (2 points), “high risk” (3–4 points), and “highest risk” (≥5 points) [34]. Moreover, Rogers Jr. et al. developed a predictive model of VTE through a logistic regression analysis of data obtained from the Patient Safety in Surgery (PSS) study, thus termed as the Rogers RAM. To do this, individual risk factors are summed and patients are divided into three categories accordingly: “low risk” (1–6 points), “moderate risk” (7–10 points), and “high risk” (>10 points) [35]. Blom et al. conducted a prospective study to estimate the incidence of VTE in 66,329 patients with cancer, thus reporting that it was 12.4/1000 patients within 6 months since the diagnosis of cancer and it was relatively higher when compared to normal healthy individuals [36]. Both thrombin formation arising from the pro-coagulant effects of tumor cells and venous compression leading to stasis are inevitable in patients with cancer [37,38]. Furthermore, cancer-treatment related factors, such as prolonged treatment period, immobilization, radiotherapy, and chemotherapy, may also raise the risk of thromboembolic events in patients with cancer [39].

The deep veins distributed in the lower extremities are classified into two categories: the proximal (the IVC, iliac, femoral, and popliteal veins) and the distal territory (the anterior, posterior tibial, peroneal, and muscular veins). Of note, proximal DVT is more frequently associated with PE and recurrence when compared to the distal one [40,41]. Our results showed that DVT occurred most frequently (31.3%) in the iliac vein. In association with this, Kahn et al. showed that thrombus was present in the common femoral vein and/or iliac vein in 25% of patients with symptomatic DVT of the lower extremities [42].

Our results showed that advanced-stage cancer was significantly more prevalent in the DVT group as compared to the non-DVT one. Presumably, this might be due to an increased immobility of the patients with advanced-stage cancer, which is in agreement with a previous report showing that patients with advanced-stage cancer are at increased risks of developing DVT [43].

In the current study, distant organ metastasis was significantly more prevalent in the DVT group as compared to the non-DVT one, as previously described [44]. But there was no significant difference in the incidence of regional lymph node metastasis between the two groups, which is not in agreement with a previous published study [45]. Involvement of metastases in the risk of DVT has been well described in the literature. Higher risks of DVT in association with distant or regional lymph node metastases may be explained by the process of metastatic dissemination. That is, there is a close association between the presence of metastasis and increased hypercoagulability because the hemostatic system might play an important role in metastatic capacity in malignancies. In more detail, intrusion of tumor cells into the blood or lymphatic fluid is an essential factor for distant metastasis. Distant metastasis is therefore followed by the interaction between tumor cells and the hemostatic system. This leads to the speculation that hypercoagulabulability may already exist in patients with regional spread of cancer [44,45].

We found that the presence of DVT had no significant correlation with the types of treatment modalities. This is in agreement with previous reports describing a lack of statistical significance in it [46,47].

In our series, despite a lack of statistical significance, the duration of cancer was shorter in the DVT group as compared to the non-DVT group. But this is not in agreement with previous published studies showing that risk factors of developing VTE include anatomical sites, biological characteristics, stage, and duration of cancer [48,49].

Lymphadenectomy is commonly used not only to assess lymph node status and the stage of gynecologic malignancies, but also to treat patients with gynecologic cancer [50,51]. But it is often accompanied by complications, such as hemorrhage, hematoma, and lymphocele [52,53]. Of these, lymphocele is one of the most common postoperative complications in that it leads to the occurrence of VTE by venous compression [54,55]. According to a review of the literature, VTE occurred after lymphadectomy at an estimated incidence of 0.8–25% [56,57,58]. This is also seen in our results. We found that lymphadectomy had a significant correlation with the occurrence of VTE.

A substantial amount of blood loss increases the risk of transfusion during perioperative period, and transfusion has been shown to be associated with the postoperative occurrence of VTE in gynecologic surgeries [59,60]. Our results also showed that the amount of intraoperative blood loss was a significant predictive factor of DVT. To summarize, distant organ metastasis, advanced stage, lymphadectomy, and amount of intraoperative blood loss had a positive predictive value for the occurrence of DVT in gynecologic cancer survivors presenting LEE. But our results cannot be generalized, because we retrospectively analyzed a small series of patients at a single, secondary medical institution. The possibility of selection bias could not therefore be completely ruled out. Further large-scale, multi-center studies are therefore warranted to establish our results.

## 5. Conclusions

It is such a potentially life-threatening condition that more than 90% of total patients with DVT of the lower extremities develop PE [61]. Clinicians should consider the possibility of LEE arising from DVT in these cases, which should be confirmed with imaging modalities such as CT venography.

## Figures and Tables

**Figure 1 healthcare-08-00048-f001:**
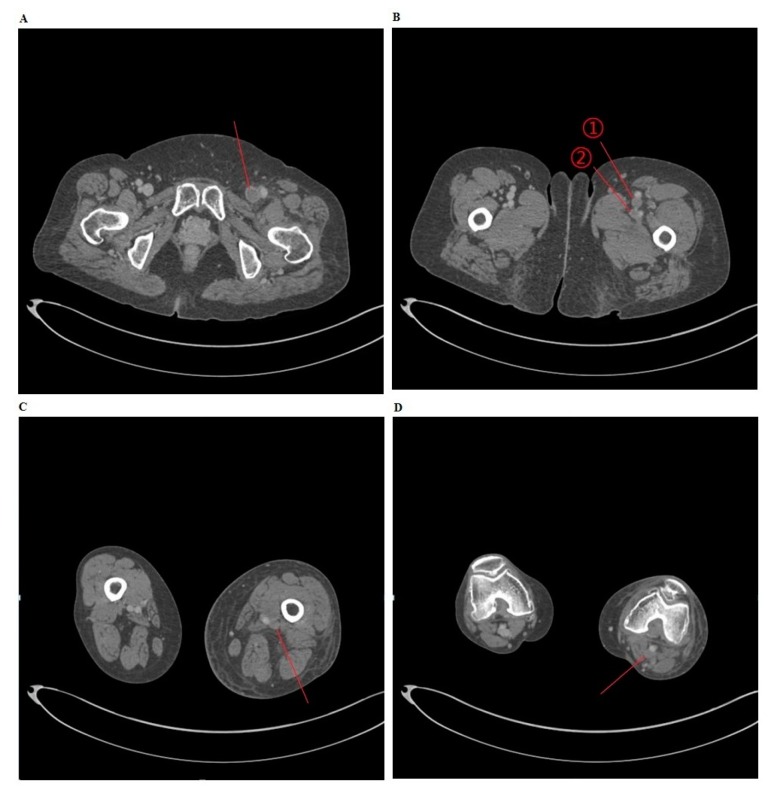
Deep vein thrombosis (DVT) on computed tomography (CT) venography.

**Figure 2 healthcare-08-00048-f002:**
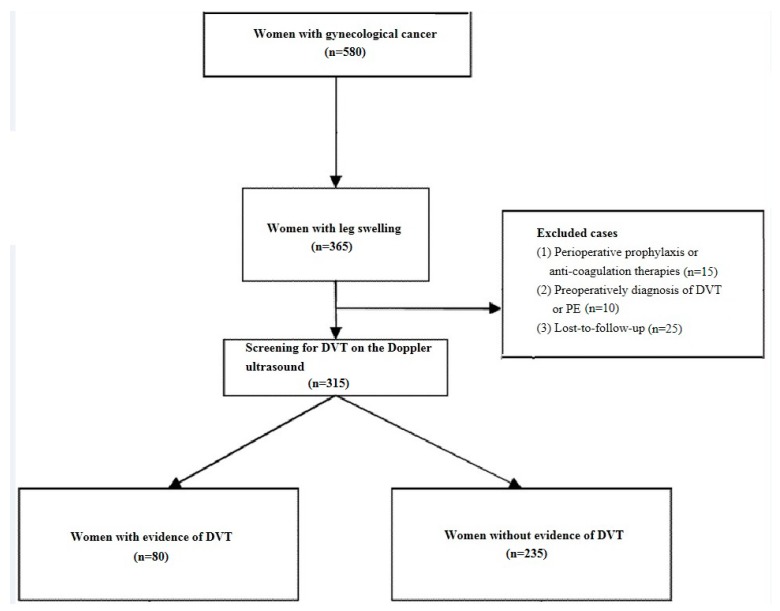
The study flow chart. **Note:** DVT, deep vein thrombosis; PE, pulmonary embolism.

**Table 1 healthcare-08-00048-t001:** Baseline characteristics of the patients.

Variables	Values
Age (years)	55.7 ± 10.4
BMI (kg/m^2^)	24.9 ± 5.3
Type of gynecologic cancer	
Cervical cancer	135 (42.9%)
Endometrial cancer	60 (19.0%)
Ovarian cancer	120 (38.1%)
TNM stage	
I	150 (47.6%)
II	35 (11.1%)
III	60 (19.0%)
IV	70 (22.3%)
Disease duration (months)	39.7 ± 26.6

**Abbreviations:** BMI, body mass index. Values are mean ± standard deviation or the number of the patients with percentage, where appropriate.

**Table 2 healthcare-08-00048-t002:** Patient characteristics in each group.

Variables	Values		*p*-value
	DVT Group (*n* = 80)	Non-DVT Group (*n* = 235)	
Age (years)	55.4 ± 11.1	56.3 ± 10.9	0.88
BMI (kg/m^2^)	26.9 ± 4.8	24.1 ± 2.7	<0.05 *
Type of gynecologic cancer			0.76
Cervical cancer	35 (43.8%)	100 (42.6%)	
Endometrial cancer	15 (18.8%)	45 (19.1%)	
Ovarian cancer	30 (37.5%)	90 (38.3%)	
TNM stage			<0.01 *
I	15 (18.8%)	135 (57.4%)	
II	15 (18.8%)	20 (8.5%)	
III	30 (37.5%)	30 (12.8%)	
IV	20 (24.9%)	50 (21.3%)	
Disease duration (months)	28.8 ± 19.6	50.9 ± 29.7	0.43

**Abbreviations:** DVT, deep vein thrombosis; BMI, body mass index. Values are mean ± standard deviation or the number of the patients with percentage, where appropriate. * Statistical significance at *p* < 0.05 by χ^2^- or Mann-Whitney *U*-test.

**Table 3 healthcare-08-00048-t003:** Predictive factors of deep vein thrombosis.

Variables	Values		OR	*p*-value
	DVT Group (*n* = 80)	Non-DVT Group (*n* = 235)		
**Metastasis**				
**Regional LN**	35 (43.8%)	75 (31.9%)	1.0	0.28
Distant organ	45 (56.3%)	80 (34.0%)	4.5	0.03 *
Treatment modalities				
CTx + RTx	20 (25.0%)	40 (17.0%)	2.5	0.37
Surgery + CTx	32 (40.0%)	105 (44.7%)	1.7	0.48
Surgery + RTx	18 (22.5%)	50 (21.3%)	3.6	0.22
Surgery + CTx + RTx	10 (12.5%)	40 (17.0%)	0.9	0.76
Co-morbidities				
DM	5 (6.3%)	20 (8.5%)	0.1	0.99
HTN	15(18.8%)	70 (29.8%)	0.4	0.54
HLD	10 (12.5%)	20 (8.5%)	0.3	0.65
AF	5 (6.3%)	20 (8.5%)	1.2	0.37
HF	5 (6.3%)	15 (6.4%)	0.8	0.41
RD	15 (18.8%)	50 (21.3%)	3.6	0.26
Advanced stage (TNM III or IV)	50 (62.5%)	80 (34.0%)	5.2	0.02 *
Circumference of the LE(cm)				
10 cm above the upper border of the patella	3.5 ± 2.9	4.3 ± 3.1	4.1	0.79
10 cm below the upper border of the patella	1.8 ± 2.1	2.7 ± 2.3	3.5	0.43
D-dimer levels (μg/L)	16.4 ± 23.9	4.6 ± 5.8	5.6	0.10
Lymphadenectomy	46 (57.5%)	106 (45.1%)	3.9	0.03 *
Operation time ≥3 hours	40 (50.0%)	139 (59.1%)	4.3	0.01 *
Amount of intraoperative blood loss ≥1500 mL	15 (18.8%)	22 (9.4%)	2.6	0.00 *
ECOG PS				
0	74 (92.5%)	215 (91.5%)	6.1	1.00
1	6 (7.5%)	20 (8.5%)		

**Abbreviations:** DVT, deep vein thrombosis; OR, odds ratio; LN, lymph node; CTx, chemotherapy; RTx, radiotherapy; DM, diabetes mellitus; HTN, hypertension; HLD, hyperlipidemia; AF, atrial fibrillation; HF, heart failure; RD, rheumatic disease; LE, lower extremity; ECOG PS, Eastern Cooperative Oncology Group performance status. Values are mean ± standard deviation or the number of the patients with percentage, where appropriate. *Statistical significance at *p* < 0.05 by χ^2^- or Mann-Whitney *U*-test.

**Table 4 healthcare-08-00048-t004:** Results of univariate and multivariate analyses of possible predictive factors.

Variables	Univariate Analysis		Multivariate Analysis	
	OR (95% CI)	*p*-value	OR (95% CI)	*p*-value
BMI	1.06 (0.71–1.57)	0.593	-	-
Distant organ metastasis	0.88 (0.67–1.26)	0.023 *	2.37 (1.98–33.76)	0.018 *
Advanced stage (TNM III or IV)	6.75 (4.27–10.11)	<0.001 *	7.15 (4.58–11.23)	<0.001 *
Lymphadenectomy	1.63 (1.09–2.23)	0.012 *	1.87 (1.19–2.86)	0.004 *
Operation time	0.71 (0.51–1.06)	0.066	-	-
Amount of intraoperative blood loss	2.25 (1.28–4.04)	0.031 *	2.04 (1.11–3.86)	0.017 *

**Abbreviations:** OR, odds ratio; CI, confidence interval; BMI, body mass index. * Statistical significance at *p* < 0.05.

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
