# Peer review of "Predictive Factors of Deep Vein Thrombosis in Gynecologic Cancer Survivors with Lower Extremity Edema: A Single-Center and Retrospective Study"

_healthcare, 2020, doi:10.3390/healthcare8010048_

Round 1

Reviewer 1 Report

The authors carried out a retrospective study to examine the predictive factors of the DVT in gynecological cancer patients. The manuscript is well-written and the study is interesting. I have few questions that the authors can answer:

By distant organ metastasis, it is safe to assume that the authors included only metastasis of lower limbs. Please clarify. To me it would make more sense if the criteria is restricted to low limb metastasis to assess the correlation with DVT.

Author Response

Point-by-point responses to the reviewer 1 comments

Dear reviewers,

We appreciate your warm comments and instructions on our manuscript.

According to your instructions, we have provided point-by-point responses as shown below.

All corresponding are highlighted in yellow changes in the manuscript.

Best regards,

Point [1] The authors carried out a retrospective study to examine the predictive factors of the DVT in gynecological cancer patients. The manuscript is well-written and the study is interesting. I have few questions that the authors can answer:

By distant organ metastasis, it is safe to assume that the authors included only metastasis of lower limbs. Please clarify. To me it would make more sense if the criteria is restricted to low limb metastasis to assess the correlation with DVT.

Response 1: We greatly appreciate your comments on our manuscript. As you have suggested, we added “Women with metastasis to the lower extremities” to inclusion criteria for the current study. (line 89)

Reviewer 2 Report

This study by Kim J et al, examined predictive factors of the deep vein thrombosis in gynecologic cancer survivors with the lower extremity edema. The results indicated retrospectively that it is necessary to consider the possibility of lower extremity edema arising from deep vein thrombosis in gynecologic cancer survivors with advanced-stage cancer as well as distant organ metastasis. The paper is straightforward, well written, and concise. Definitely deserves to be published and is a valuable contribution to the “healthcare” journal. Some minor flaws need to be addressed before publication.

Minor points:

[1] Introduction, Lines 28-30:

Please, add here a sentence about the triad of Virchow, who proposed risk factors contributing to venous thromboembolism (venous stasis, endothelial injury and hypercoagulable states).

[2] Introduction, Lines 39-41:

Please, provide data of the incidence of postoperative venous thromboembolism in ovarian cancer. This has been reported to be as high as 13.2%, even in the setting of prophylaxis.

Relevant reference: Gunderson CC, et al. The survival detriment of venous thromboembolism with epithelial ovarian cancer. Gynecol Oncol. 2014 Jul;134(1):73-7.

[3] Introduction, Lines 53-60:

There is a trend, catheter venography, to be recommended only when interventional treatment such as thrombolysis is planned because of its invasiveness. Please, make also a statement about compression vein ultrasonography with color Doppler flow or duplex ultrasonography, which is the most frequently used investigation in the diagnosis of deep vein thrombosis.

Relevant reference: Cohen A, et al. Venous Thromboembolism in Gynecological Malignancy. Int J Gynecol Cancer. 2017 Nov;27(9):1970-1978.

[4] Discussion, Line 153:

Please, add here some information about the two well known risk assessment tools, the Caprini (1) and the Rogers scores (2), respectively. Caprini score assigns points to various venous thromboembolism risk factors and each patient is categorized by his resulting score as being at low, moderate, high or very high risk of venous thromboembolism. Rogers score was developed in a population of general surgery patients using logistic regression modeling. Points are assigned to various patient and procedure risk factors. Patients are categorized into risk groups on the basis of their final scores.

Relevant references:

(1) Caprini JA. Thrombosis risk assessment as a guide to quality patient care. Dis Mon. 2005 Feb-Mar;51(2-3):70-8.

(2) Rogers SO Jr, et al. Multivariable predictors of postoperative venous thromboembolic events after general and vascular surgery: results from the patient safety in surgery study. J Am Coll Surg. 2007 Jun;204(6):1211-21.

Author Response

Point-by-point responses to the reviewer 2 comments

Dear reviewers,

We appreciate your warm comments and instructions on our manuscript.

According to your instructions, we have provided point-by-point responses as shown below.

All corresponding are red changes in the manuscript.

Best regards,

Point [1] Introduction, Lines 28-30:

Please, add here a sentence about the triad of Virchow, who proposed risk factors contributing to venous thromboembolism (venous stasis, endothelial injury and hypercoagulable states).

Response 1: I would like to express my sincere gratitude for your interest in our research and a detailed review.

“According to Virchow, there are a triad of risk factors that contribute to venous thromboembolism; these include venous stasis, endothelial injury and hypercoagulable states [3].” (line 28-29 ).

Point [2] Introduction, Lines 39-41:

Please, provide data of the incidence of postoperative venous thromboembolism in ovarian cancer. This has been reported to be as high as 13.2%, even in the setting of prophylaxis.

Relevant reference: Gunderson CC, et al. The survival detriment of venous thromboembolism with epithelial ovarian cancer. Gynecol Oncol. 2014 Jul;134(1):73-7.

Response 2: Thank you for the review that will improve the accuracy of your paper. We added the following:

“Its incidence of postoperative venous thromboembolism in ovarian cancer is reported to be 13.2%( DVT 71.6%, PE 24.3%, both 4.1%) [8].”(line 39-41).

Point [3] Introduction, Lines 53-60:

There is a trend, catheter venography, to be recommended only when interventional treatment such as thrombolysis is planned because of its invasiveness. Please, make also a statement about compression vein ultrasonography with color Doppler flow or duplex ultrasonography, which is the most frequently used investigation in the diagnosis of deep vein thrombosis.

Relevant reference: Cohen A, et al. Venous Thromboembolism in Gynecological Malignancy. Int J Gynecol Cancer. 2017 Nov;27(9):1970-1978.

Response 3: Thank you for your interest and attention in improving the completeness of the paper.

Compression vein ultrasonography is a test of increasing importance in the diagnosis of DVT.

Thank you for recommending a meaningful reference paper. An additional description of compression vein ultrasonography is provided in the revised manuscript.

“Moreover, compression vein ultrasonography with color Doppler flow or duplex ultrasonography is also recommended [22].”(line 66-67).

Point [4] Discussion, Line 153:

Please, add here some information about the two well known risk assessment tools, the Caprini (1) and the Rogers scores (2), respectively. Caprini score assigns points to various venous thromboembolism risk factors and each patient is categorized by his resulting score as being at low, moderate, high or very high risk of venous thromboembolism. Rogers score was developed in a population of general surgery patients using logistic regression modeling. Points are assigned to various patient and procedure risk factors. Patients are categorized into risk groups on the basis of their final scores.

Relevant references:

(1) Caprini JA. Thrombosis risk assessment as a guide to quality patient care. Dis Mon. 2005 Feb-Mar;51(2-3):70-8.

(2) Rogers SO Jr, et al. Multivariable predictors of postoperative venous thromboembolic events after general and vascular surgery: results from the patient safety in surgery study. J Am Coll Surg. 2007 Jun;204(6):1211-21.

Response 4: We thank you for your careful attention. The following text was added to the revised manuscript.

“Such risk factors are evaluated based on two well-known instruments. The Caprini risk assessment model (RAM), originally developed for surgical patients, aims to promote the derivation of risk factors of developing VTE. To do this, individual risk factors are summed and patients are divided into four categories accordingly: “low risk” (0-1 points), “moderate risk” (2 points), “high risk” (3-4 points) and “highest risk” (≥5 points) [34]. Moreover, the Rogers Jr. et al. developed a predictive model of VTE through a logistic regression analysis of data obtained from the Patient Safety in Surgery (PSS) study, thus termed as the Rogers RAM. To do this, individual risk factors are summed and patients are divided into three categories accordingly: “low risk” (1-6 points), “moderate risk” (7-10 points), “high risk” (>10 points) [35].” (line 183-192 ).

Reviewer 3 Report

The Idea of this work is fine. The large database they have is a great basis. However, I do not see the novelty or additional information in this paper. There are by far larger studies.

However, they specifically focus on gynecological studies which might make it more special. So, there might be a potential interest in this paper. 

A few comments for optimization:

The introduction should be more focussed on the main aims/results of this study. The information on diagnosis is not really helpful. The authors should emphasize the clinical meaning of thrombosis and why better knowledge on predictive factors could be helpful

Methods: The results of this paper are all based on statistics. Please elaborate on that. Did you have missing values? how did you handle it? Have you considered multivariate analyses?

Results:

I miss a few important varaibes: ECOG, type of surgery (esp. lymph node dissection), length of surgery , blood loss, length of hospitalization ....

These factors must be taken into account.

Additionally, multivariate analyses have to be performed to allow for any conclusions.

Author Response

Point-by-point responses to the reviewer 3 comments

Dear reviewers,

We appreciate your warm comments and instructions on our manuscript.

According to your instructions, we have provided point-by-point responses as shown below.

All corresponding are highlighted in blue changes in the manuscript.

Best regards,

Comments and Suggestions for Authors

The Idea of this work is fine. The large database they have is a great basis. However, I do not see the novelty or additional information in this paper. There are by far larger studies.

However, they specifically focus on gynecological studies which might make it more special. So, there might be a potential interest in this paper.

A few comments for optimization:

Point [1] The introduction should be more focused on the main aims/results of this study. The information on diagnosis is not really helpful. The authors should emphasize the clinical meaning of thrombosis and why better knowledge on predictive factors could be helpful

Response 1: We greatly appreciate your instructions on our manuscript. We deleted the following statements:

“Patients with gynecologic malignancy are also at increased risk of developing VTE, for which advanced stages, pelvic masses and lengthy abdominal and pelvic operations serve as risk factors [7,8]. Typical manifestations of DVT include signs and symptoms, such as lower extremity edema (LEE), tenderness, pain, warmth or red discoloration [9].”

To clarify the background and rationale of the current study, we added the following statements to the Introduction:

“In more detail, without prophylaxis, the incidence of VTE is estimated at approximately 10-40% [9]. With prophylaxis, it is estimated at 1.14% in patients diagnosed with gynecological disease, 0.7% in those undergoing laparoscopic gynecological surgery, 0.3% in those undergoing urogynecological surgery and 4% in those with gynecological malignancies [10-13]. It remains problematic, however, that most of published studies evaluated symptomatic cases rather than asymptomatic ones; the latter could be frequently neglected without efficient methods for detecting VTE. Indeed, approximately 50% of total patients with VTE are presumed to be silent cases [14]. It can therefore be inferred that the actual incidence of postoperative VTE might be higher as compared with published reports [14].”(line 41-49)

“According to the guidelines from American College of Chest Physicians (ACCP) and American College of Obstetricians and Gynecologists (ACOG), appropriate preventive interventions are recommended based on diverse levels of postoperative risks of developing VTE [9,23]. Nevertheless, there is a paucity of evidence that advocates the necessity to stratify patients undergoing gynecological surgery according to the level of risks of developing VTE [24-26]. In these patients, several postoperative risk factors of developing VTE have been suggested; these include body mass index (BMI) 30 or 40 kg/m2, operation time >180 minutes, cancer surgery and blood transfusion 2000 mL [19,27,28]. But this cannot be generalized because the corresponding studies failed to efficiently assess risk factors of developing VTE.”(line 68-76)

Point [2] Methods: The results of this paper are all based on statistics. Please elaborate on that. Did you have missing values? how did you handle it? Have you considered multivariate analyses?

Response 2: We greatly appreciate your instructions on our manuscript. According to your instructions, we performed a rigorous review of the study design and statistical methods. We have now noticed that it is mandatory to perform a multivariate analysis of the patient data before drawing conclusions. We therefore added the following statements:

“Furthermore, univariate and multiple logistic regression analyses were also performed to identify significant correlations between risk factors of developing DVT and adjusted or unadjusted variables. Their results were expressed as odds ratios (ORs) with 95% confidence intervals (95% CIs).”(line 125-128)

Point [3] Results:

I miss a few important variables: ECOG, type of surgery (esp. lymph node dissection), length of surgery, blood loss, length of hospitalization....

These factors must be taken into account.

Additionally, multivariate analyses have to be performed to allow for any conclusions.

Response 3: We greatly appreciate your instructions on our manuscript. As we responded to your above comments, we added the following statements to clarify the background the current study:

“According to the guidelines from American College of Chest Physicians (ACCP) and American College of Obstetricians and Gynecologists (ACOG), appropriate preventive interventions are recommended based on diverse levels of postoperative risks of developing VTE [9,23]. Nevertheless, there is a paucity of evidence that advocates the necessity to stratify patients undergoing gynecological surgery according to the level of risks of developing VTE [24-26]. In these patients, several postoperative risk factors of developing VTE have been suggested; these include body mass index (BMI) 30 or 40 kg/m2, operation time >180 minutes, cancer surgery and blood transfusion 2000 mL [19,27,28]. But this cannot be generalized because the corresponding studies failed to efficiently assess risk factors of developing VTE.”(line 68-76)

We have now noticed that it is mandatory to perform a multivariate analysis of the patient data before drawing conclusions. We therefore added the following statements:

“Furthermore, univariate and multiple logistic regression analyses were also performed to identify significant correlations between risk factors of developing DVT and adjusted or unadjusted variables. Their results were expressed as odds ratios (ORs) with 95% confidence intervals (95% CIs).” (line 125-128)

We added the following statement to the inclusion criteria for the current study:

“(5) Women with Eastern Cooperative Oncology Group (ECOG) performance status of 0 or 1 [29].” (line 90)

We revised the Results:

“But the distal organ metastasis, advanced-stage cancer, lymphadectomy, operation time ³3 hours and amount of intraoperative blood loss ³1,500 mL were significantly more prevalent in the DVT group as compared with the non-DVT group (p<0.05).”(line 155-157)

3.4. Results of univariate and multivariate analyses of possible predictive factors

We performed both univariate and multivariate analyses of predictive factors, such as BMI, distant organ metastasis, advanced stage, lymphadectomy, operation time ³3 hours and amount of intraoperative blood loss ³1,500 mL, showing a significant difference between the DVT group and the non-DVT group. This showed that distant organ metastasis, advanced stage, lymphadectomy and amount of intraoperative blood loss ³1,500 mL were found to be significant predictive factors (Table 4).”(line 165-171)

Table 4. Results of univariate and multivariate analyses of possible predictive factors.

Variables

Univariate analysis

Multivariate analysis

OR

(95% CI)

p-value

OR

(95% CI)

p-value

BMI

1.06 (0.71-1.57)

0.593

-

-

Distant organ metastasis

0.88 (0.67-1.26)

0.023*

2.37 (1.98-33.76)

0.018*

Advanced stage

(TNM III or IV)

6.75 (4.27-10.11)

<0.001*

7.15 (4.58-11.23)

<0.001*

Lymphadenectomy

1.63 (1.09-2.23)

0.012*

1.87 (1.19-2.86)

0.004*

Operation time

0.71 (0.51-1.06)

0.066

-

-

Amount of intraoperative blood loss

2.25 (1.28-4.04)

0.031*

2.04 (1.11-3.86)

0.017*

Abbreviations: OR, odds ratio; CI, confidence interval; BMI, body mass index.

*Statistical significance at p<0.05.

We revised the Discussion as follows:

“Lymphadenectomy is commonly used not only to assess lymph node status and the stage of gynecologic malignancies but also to treat patients with gynecologic cancer [52,53]. But it is often accompanied by complications, such as hemorrhage, hematoma and lymphocele [54,55]. Of these, lymphocele is one of the most common postoperative complications in that it leads to the occurrence of VTE by venous compression [56,57]. According to a review of literatures, the VTE occurred after lymphadectomy at an estimated incidence of 0.8-25% [58-60]. This is also seen in our results; we found that lymphadectomy had a significant correlation with the occurrence of VTE.”(line 230-236)

“A substantial amount of blood loss increases the risk of transfusion during perioperative period, and transfusion has been shown to be associated with the postoperative occurrence of VTE in gynecologic surgeries [61,62]. Our results also showed that the amount of intraoperative blood loss was a significant predictive factor of DVT. To summarize, distant organ metastasis, advanced stage, lymphadectomy and amount of intraoperative blood loss had a positive predictive value for the occurrence of DVT in gynecologic cancer survivors presenting with LEE.”(line 237-242)

Round 2

Reviewer 1 Report

The authors addressed my concerns and I support this study for publication.